# Correcting for Intra-Individual Variability in Sodium Excretion in Spot Urine Samples Does Not Improve the Ability to Predict 24 h Urinary Sodium Excretion

**DOI:** 10.3390/nu12072026

**Published:** 2020-07-08

**Authors:** Karen Elizabeth Charlton, Aletta Elisabeth Schutte, Leanda Wepener, Barbara Corso, Paul Kowal, Lisa Jayne Ware

**Affiliations:** 1School of Medicine, University of Wollongong, Wollongong 2522, Australia; 2Illawarra Health and Medical Research Institute, Wollongong 2522, Australia; 3School of Public Health and Community Medicine, University of New South Wales, The George Institute for Global Health, Sydney 2052, Australia; a.schutte@unsw.edu.au; 4Hypertension in Africa Research Team (HART), North-West University, Potchefstroom 2520, South Africa; leandawepener@gmail.com; 5Neuroscience Institute, National Research Council, 35121 Padova, Italy; barbara.corso@in.cnr.it; 6Research Institute for Health Sciences, Chiang Mai University, Chiang Mai 50200, Thailand; paul.r.kowal@gmail.com; 7World Health Organization (WHO), CH-1211 Geneva 27, Switzerland; 8SA MRC/Wits Developmental Pathways for Health Research Unit, University of the Witwatersrand, Johannesburg 2013, South Africa; lisa.ware@wits.ac.za

**Keywords:** sodium, salt, urinary excretion, spot urine, equations, variability, sensitivity

## Abstract

Given a global focus on salt reduction efforts to reduce cardiovascular risk, it is important to obtain accurate measures of salt intake on a population level. This study determined firstly whether adjustment for intra-individual variation in urinary sodium (Na) excretion using three repeated 24 h collections affects daily estimates and whether the use of repeated spot urine samples results in better prediction of 24 h Na compared to a single collection. Twenty three community-dwelling men and women from South Africa (mean age 59.7 years (SD = 15.6)) participating in the World Health Organization Study on global AGEing and adult health (WHO-SAGE) Wave 3 study collected 24 h and spot early morning urine samples over three consecutive days to assess urinary Na excretion. INTERSALT, Tanaka, and Kawasaki prediction equations, with either average or adjusted spot Na values, were used to estimate 24 h Na and compared these against measured 24 h urinary Na. Adjustment was performed by using the ratio of between-person (sb) and total (sobs) variability obtained from repeated measures analysis of variance. Sensitivity of the equations to predict daily urinary Na values below 5 g salt equivalent was calculated. The sb/sobs for urinary Na using three repeated samples for spot and 24 h samples were 0.706 and 0.798, respectively. Correction using analysis of variance for 3 × 24 h collections resulted in contraction of the upper end of the distribution curve (90th centile: 157 to 136 mmoL/day; 95th centile: 220 to 178 mmoL/day). All three prediction equations grossly over-estimated 24 h urinary Na excretion, regardless of whether a single spot urine or repeated collections corrected for intra-individual variation were used. Sensitivity of equations to detect salt intake equivalent values of ≤5 g/day was 13% for INTERSALT, while the other two equations had zero sensitivity. Correcting for intra-individual variability in Na excretion using three 24 h urine collections contracted the distribution curve for high intakes. Repeated collection of spot samples for urinary Na analysis does not improve the accuracy of predicting 24 h Na excretion. Spot urine samples are not appropriate to detect participants with salt intakes below the recommended 5 g/day.

## 1. Introduction

Many leading medical and public health organizations recommend reducing dietary sodium (Na) to a maximum of 2000 mg per day (5 g salt) [1] on the basis of evidence indicating a public health benefit [2]. There is strong evidence of a linear, dose–response effect of sodium reduction on blood pressure (BP) [3]. In addition, the evidence shows that sodium reduction prevents cardiovascular disease [4].

In order to monitor the effectiveness of salt reduction strategies on a population level, it is important to collect regular and representative indicators of habitual salt intake. Multiple, non-consecutive, 24 h urine collections are the gold standard for assessing sodium intake [5,6]. Using the method of repeated 24 h urine collections over 1.5 to 4 years, the Trials of Hypertension Prevention (TOHP) study demonstrated a linear 17% increase in cardiovascular disease risk per 1000 mg/day increase in sodium, from levels starting as low as 1500 mg/day among 2275 participants [7].

However, the high burden of labour and difficulty to collect complete 24 h urine samples has prompted the search for more practical alternative methods, such as spot urinary Na concentrations. For this purpose, a number of prediction equations that include spot Na and creatinine concentrations have been developed by multiple research groups. Of these, the most commonly used are INTERSALT [8], Kawasaki [9], and Tanaka [10]. These equations have been validated against 24 h urinary sodium excretion, but mostly in American or Japanese populations. Authors have reported that these equations unacceptably overestimate individual intake at low levels and underestimate intake at high levels, even while being unbiased at the average level [5,11].

Collection of 24 h urine samples is particularly onerous in low-middle income countries (LMICs) where poor infrastructure exists for transport and laboratory services, especially in rural areas. The application of equations to spot urine samples in order to estimate 24 h excretion concentrations has been used in some large observational studies in LMICs, such as the Prospective Urban Rural Epidemiology (PURE) study [12], but this has raised much criticism [13,14] regarding potentially erroneous conclusions on the association between salt intake and cardiovascular disease outcomes. We have previously reported that equations using spot urine collections have an unacceptably wide degree of bias and are not appropriate for use in the South African population with Tanaka, Kawasaki, and Mage equations—all overestimating salt intake (15). The INTERSALT equation systematically underestimated measured 24 h sodium excretion [15]. Other authors have reported similar findings [16].

Intraperson variability in sodium intake can be as great or greater than interpersonal variability due to the varied nature of food intake from day-to-day [5,17]. Because of measurement challenges and day-to-day variability in sodium intake, estimating population averages is subject to far less error than estimating individual intake. A method to reduce or remove the effects of measurement error due to intra-individual variation would allow a more accurate description of habitual salt intake. Several methods exist to correct for intra-individual variation in population surveys. One method is to collect multiple collections for each participant and average their data. Another method is to apply a correction factor to the distribution [18,19]. This requires estimating the correction factor, for example, by collecting multiple samples from a representative subset of the survey population. The correction method has been applied many times to dietary intake data [20,21,22] but less frequently to biochemical data [23,24,25]. We have previously applied this method to spot urinary iodine concentrations (UIC) in a sample of older Australians and found that it narrowed the extremes of UIC at the upper end of the curve [26].

As a population reduces salt intake, accurate estimation becomes more important for refining intervention efforts and monitoring success [27,28]. The objective of this study was to evaluate the impact of collecting three consecutive repeated 24 h samples for calculating the correction factor for urinary Na excretion and comparing this with the effect of averaging the results for each person. It also assesses whether the use of three repeated spot samples to correct spot urinary Na concentrations in prediction equations improves the sensitivity and specificity to classify individuals with salt intakes above and below the recommended 5 g salt/day cut off.

## 2. Materials and Methods

Data was collected in a sub sample of participants included in the Salt and Tobacco World Health Organization Study on global AGEing and adult health (WHO-SAGE) study in South Africa. SAGE is a multinational study examining the health and well-being of adult populations and the ageing process [29]. Evaluation of the health effects of the mandatory salt reduction policy [30] on South African adults is being conducted using a nested study design in waves 2 and 3 [31]. Inclusion criteria for urine collection were: respondent must be part of the WHO-SAGE cohort, with no indication of urinary incontinence or another condition that could impede 24 h urine collection; and if female, not menstruating, pregnant, or breast feeding on the day of collection.

Survey teams were trained with support from WHO Geneva, with survey teams using standardised household, individual, and proxy questionnaires, anthropometry, blood sampling, BP, and physical function tests as described previously in SAGE wave 1 [29]. The study protocol used for sodium determination in 24 h urine samples followed the WHO/Pan American Health Organization (PAHO) guidelines [27]. Respondents were requested to collect all urine produced for 24 h, excluding the first pass urine on day 1, but including the first urine of the following morning (day 2) in a 5 L plastic container using 1 g thymol as a preservative. Spot samples were collected without preservative from the second urine passed on day 1 (marking the start of the 24 h collection) and decanted into three 15 mL Porvair tubes (Porvair Sciences, Leatherhead, UK), then kept in a thermoelectric cooler box powered by the fieldwork vehicles and containing ice packs to maintain a lowered temperature. Early morning rather than random samples were collected for spot samples due to logistics of the fieldwork and because of the evidence that early morning samples more closely reflect diurnal variation [32,33]. Each 24 h sample was collected the next morning, total volumes were recorded, and aliquots generated with all samples then shipped to the laboratory, maintaining the cold chain using precooled ice packs as a means to maintain temperature control. When the samples arrived at the laboratory, the cooler box was examined and the temperature of the samples noted and recorded. Thymol preservative, a crystalline natural derivative of the thyme plant, has been shown not to affect urinary creatinine, sodium, or potassium concentrations for up to 5 days after collection at room temperature [34].

Incomplete 24 h urine collections were assumed if: total volume <300 mL; or creatinine excretion <4 mmoL/day (women) or <6 mmoL/day (men) [35]. Sodium was determined using the indirect ion-selective electrode method and creatinine analysed using the standardised urinary Jaffe kinetic method (Beckman Coulter Synchron DXC600/800 System). For BP measurements, validated wrist-worn Omron BP devices (R6, Omron, Japan) [34] were used to record three sequential measures on the left arm (1 min between each measure and following 5 min at rest). The participant held their wrist directly at the level of the heart using inbuilt positional sensors in the device and was seated with legs uncrossed throughout the measurements. The mean of the second and third readings were used to generate a blood pressure value. Blood pressure values were deemed as valid if: Systolic (Sys) > Diastolic (Diast); 80 ≤ Sys ≤ 270 mmHg; 40 ≤ Diast ≤ 180 mmHg; and pulse pressure (Sys-Diast) ≥ 13 [36].

### Statistical Analysis

Data were transformed using the natural logarithm to improve normality. Repeated measures analysis of variance was performed to determine the between person (sb) and total (sobs) variability [18,19]. An adjusted log Na (for spot and 24 h) value was calculated for each person as: Adjusted Na (spot or 24 h) = ((person’s day 1 Na− group mean for day 1) × (sb/sobs)) + group mean for day 1.

The results were exponentiated. First, the correction factor (sb/sobs) was calculated using only the first two replicates, then it was calculated using all three replicates. This was done for spot Na measures (mmoL/L) and also for 24 h Na concentrations (mmoL/day). The average for each person was calculated using the first two day replicates and for all three days. Centiles of the distribution were calculated for the raw day 1 data and for distributions derived using adjustment or averaging.

In order to compare urinary Na estimates between spot and 24 h collections, the INTERSALT, Tanaka, and Kawasaki equations were applied to spot Na values, using single, 2 day average, 3 day average, and 2 and 3 day corrected values in the equations. In the case of Tanaka and Kawasaki equations, spot creatinine was similarly averaged or adjusted (Appendix A). The resultant predicted 24 h values were compared with the first 24 h Na collection. Correlations were conducted by Spearman coefficients and the sensitivity of the predicted proportion below and above 5 g salt per day estimated. Wilcoxon signed rank test was applied to test the difference between measured 24 h Na to the estimation obtained by the implementation of the three different equations. Analyses were performed using STATA SE, version 12.1 (Stata Statistical Software: Release 12. College Station, StataCorp LP, TX, USA).

SAGE was approved by the WHO Ethics Review Committee (reference number RPC149) with local approval from the North-West University Human Research Ethics Committee and University of the Witwatersrand Human Research Ethics Committee. All respondents provided written informed consent prior to taking part in the study. The study complies with the ethical principles for medical research involving human participants as per the Declaration of Helsinki.

## 3. Results

Of the 48 participants who collected repeated urine samples, only 23 had three days of valid collections and were thus included in the analysis. These were all from the same ethnic background (African/Black). Descriptive data is presented as median (IQR) because of non normal distributions. Background characteristics are shown in Table 1, while the median Na, K, and Cr values in spot and 24 h urine, by collection day, are reported in Table 2.

### 3.1. Distribution of Repeated Spot and 24 h Urinary Na Concentrations

The distribution of spot and 24 h urinary Na concentration is shown in Table 3, according to raw data from Day 1 only, after correction for intra-individual variation of two and three spot sample collections and for averages of multiple collections. This is graphically represented for spot and 24 h urinary Na concentrations in Figure 1; Figure 2, respectively.

Despite the median being similar for Day 1 and corrected 3-day values for 24 h samples, the upper end of the distribution was contracted (90th (p90) and 95th (p95) percentile reduced from 201 and 220 mmoL/day, respectively, to 170 and 182 mmoL/day) (Table 3). In the case of spot urinary Na, correction for 3 days reduced the p90 and p95 from 163 and 186, respectively, to 125 and 135 mmoL/L, respectively. The proportion of participants categorised as having salt intake equivalents ≤5g/day, according to a single sample, was 39.1%, which remained the same for an average of days 1 and 2, as well as values adjusted for either days 1 and 2 or adjusted for all 3 days, but was slightly lower when all 3 days were averaged (34.8%).

Spearman’s correlations for log-transformed urinary 24 h Na (mmoL/day) concentration values were: days 1 and 2: *r* = 0.80, *p* < 0.0001; days 1 and 3: *r* = 0.53, *p* = 0.0086; days 2 and 3: *r* = 0.55, *p* = 0.0060. While the correlations for log-transformed urinary spot Na (mmoL/L) concentration values were: days 1 and 2: *r* = 0.29, *p* = 0.1812; days 1 and 3: *r* = 0.52, *p* = 0.0117; days 2 and 3: *r* = 0.44, *p* = 0.0364. Similarly, spot potassium (K) and creatinine (Cr) urinary concentration distribution is shown according to raw data collected on day 1 and after adjustment for intra-individual variation of two and three spot sample collections and averages of multiple collections in Appendix A.

### 3.2. Prediction Equations to Estimate 24 h Na from Spot Na

Of the three prediction equations, the INTERSALT prediction equation most closely approximated measured 24 h Na excretion (day 1), however values were still grossly over-estimated (Table 4). Compared to a median 24 h Na excretion of 107 mmoL/day, using a single (Day 1) spot urine value in the equation resulted in an estimate of 133–141 mmoL/day, which was similar regardless of whether the average of 2 or 3 repeated spot concentrations was used in the equation or whether correction factors for intra-individual variability were applied.

The sensitivity of the INTERSALT equations to classify subjects according to 24 h Na excretion equivalent to ≤5 g salt per day was 11.1%, while the specificity (classification of intakes >5 g salt equivalent/day) was 92.9%. Sensitivity and specificity remained the same within all five estimates (single day, mean, or corrected values for days 1 and 2 or for all 3 days). Using a single spot sample in the INTERSALT equation resulted in 52% subjects having a relative difference of >40% compared to measured 24 h Na and only 4.4% subjects within 10%. The relative difference >40% or <10% for the other values were, respectively, 56.5% and 13.0% for mean days 1 and 2; 60.9% and 8.7% for adjusted days 1 and 2; 56.5% and 4.4% for the mean value of all 3 days; and 60.9% and 8.7% for adjusted for all 3 days (data not shown).

Using the INTERSALT equation, compared to day 1 for 24 h urine collection, the proportion of participants with salt equivalent intakes ≤5 g/day (39.1%) was 8.7% for spot day 1 alone and this remained unchanged for any further day collections, expressed either as averages or adjusted values over 2 or 3 days.

Both Tanaka (Table 5) and Kawasaki (Table 6) equations performed poorly, whether a single (Day 1) spot sample was used or means and adjusted variations of 2 and 3 day collections, with zero sensitivity (no subjects classified as ≤5 g salt equivalent/day) and specificity of 100% for >5 g salt/day across all iterations. In other words, neither equation categorised any participants as having salt intake equivalents of ≤5 g/day. Using the Tanaka equations, 95.7% of subjects were classified as having >40% difference between measured and predicted 24 h Na excretion, regardless of whether single or repeated spot urinary Na concentrations (corrected or averaged) were used in the equations. Whereas when the Kawasaki equation was applied, all subjects were classified as having >40% difference between measured and predicted 24 h Na excretion.

## 4. Discussion

In a sample of older South African adults, correcting for intra-individual variability in daily salt intake using three repeated 24 h urine collections did not alter median Na values but did contract the upper tail of the distribution curve to result in lower values in the uppermost percentiles. The similarity of 24 h Na excretion across three days of urine collection in the current study may reflect a less varied diet than that reported in other populations.

To our knowledge, this is the first study to consider whether repeated spot urine collections could improve the validity of prediction equations to estimate 24 h salt intake. We have confirmed that the use of three repeated spot urinary Na concentrations, corrected for intra-individual variability, did not result in more accurate predicted 24 h Na excretion values. Our data does not support the use of either a single or two or three repeated spot urinary Na concentrations to estimate habitual salt intake. Of the three prediction equations applied to the dataset, all grossly over estimated 24 h urinary Na excretion, with INTERSALT performing somewhat better.

Many studies have shown that spot urine samples have limited applicability in determining 24 h salt intake at a population level. In our study, all spot collections were taken as early morning samples, as has been done in another validation study that found three repeated early morning spot samples most closely approximated 24 h urinary Na collections [37]. However, other studies have reported that higher numbers of repeated spot urine collections, ranging from four to seven collected at random, improved accuracy of estimates of repeated 24 h urinary excretions [38,39]. A validation study in Chinese adults found that all three equations (Kawasaki, INTERSALT, and Tanaka) performed poorly in estimating 24 h urinary sodium excretion [40]. In that study, the Kawasaki formula provided the least biased estimate of sodium intakes, while the INTERSALT formula showed the largest bias. The PURE (Prospective Urban Rural Epidemiology)-China Study reported similar findings [41]. However, in the current study, the INTERSALT equations provided the least-biased information, as was reported in an American population aged 18–39 years [32].

The Kawasaki and Tanaka formulae were originally developed in Japanese populations [9,10] that have very high salt intakes, whereas the INTERSALT formula was based on Western populations [8]. The finding that the Kawasaki equation provided estimates six times that 24 h urinary Na excretion is in contrast to other research [42], which suggests that while spot urine estimates under- and over-estimate actual excretion at extremes of the range, mean levels are relatively close. Another study from South Africa reported a lower magnitude of error but similarly reported the greatest bias with the Kawasaki equation and least with the INTERSALT equation [16]. Conflicting findings may thus be due to differences in dietary patterns and confounding factors associated with ethnicity, culture, cuisine, as well as differences in body size and composition. One study by Nguyen et al. (2019) applied the Tanaka formula to annual urine spot samples taken in Japanese workers (*n* = 4360) over a five year period [43]. They showed that for each SD increase in estimated 24 h sodium, there were small but significant increases in both systolic and diastolic BP. Many studies have confirmed the relationship between vascular health and Na intake, including those with ultra-precise methodology such as long-term Na balance studies [44]. A highly controlled study suggested that compared to a single 24 h collection, three consecutive 24 h urine collections improved the precision to predict dietary Na intake from 49% to 75% accuracy and for K from 66% to 81% accuracy [45]. While the results obtained at controlled fixed intakes can demonstrate inherent variability in the measures, they are not reflective of people’s daily lives, which introduce further variability from differences in daily food intakes, hydration status, temperature, humidity, and exercise levels. As such, even multiple 24 h urine collections are likely to be, at best, an estimate of habitual intake, albeit with a greater level of accuracy. Previous research has suggested that up to 10 repeated 24 h urine samples may be required for an accurate estimate of usual sodium intake [46,47]. Given this and the many publications in the literature using sodium estimates from spot urine samples, interpretation of the findings of such studies should be viewed with caution. Given the older age group and overall low 24 h Na excretion, generalizability to wider age groups that may have more diverse dietary patterns is limited.

Our study is limited by a small sample size, yet we successfully obtained both 24 h and spot urine samples in a consistent manner. These samples allowed unique comparisons of intra-individual 24 h Na and spot Na concentrations over 3 days and the accuracy in using different formulae to determine daily salt intake in a general population sample with similar habits and within the same region.

## 5. Conclusions

Accurate measurement of population-level salt intake is required to monitor progress toward salt-reduction targets, but collection of 24 h urine samples to analyse Na is burdensome. Spot urine collections, proposed as a proxy for assessing 24 h Na excretion using prediction equations, are not valid for use in an urban older obese population in South Africa. Correction for intra-individual variability using three repeated spot urine collections did not improve validity, nor sensitivity in characterizing individuals with low salt intakes. The use of three repeated 24 h urinary Na collections resulted in a shrinkage of population distribution at the upper extremes but did not impact on overall assessment of the median. This data suggests that a single 24 h urinary Na collection is appropriate for use in population surveys to detect habitual salt intake.

## Figures and Tables

**Figure 1 nutrients-12-02026-f001:**
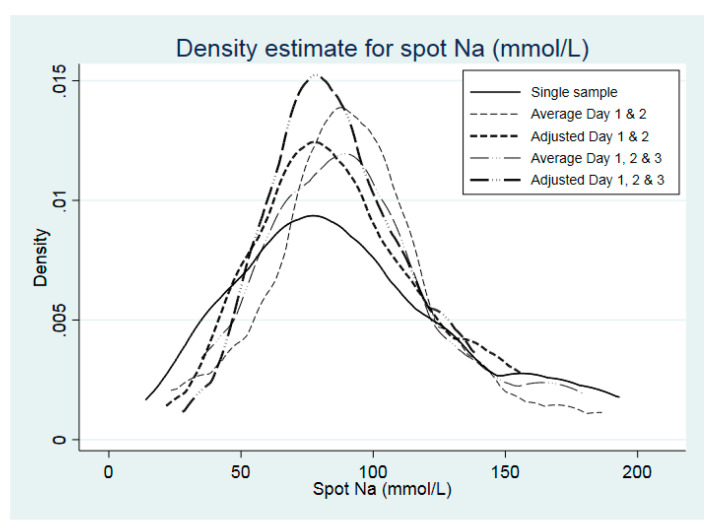
Population distribution of urinary spot Na (mmoL/L), according to number of days of urine collection (*n* = 23). Density refers to proportion of persons, e.g. 0.005 = 0.5%). Solid line: raw data from day1 only; thin dashed line: after correction for average of day1 and day2; thick dashed line after correction for intra-individual variation of day1 and day2; thin long dashed & dotted line: after correction for average of all three days; thick long dashed & dotted line after correction for intra-individual variation of all three days.

**Figure 2 nutrients-12-02026-f002:**
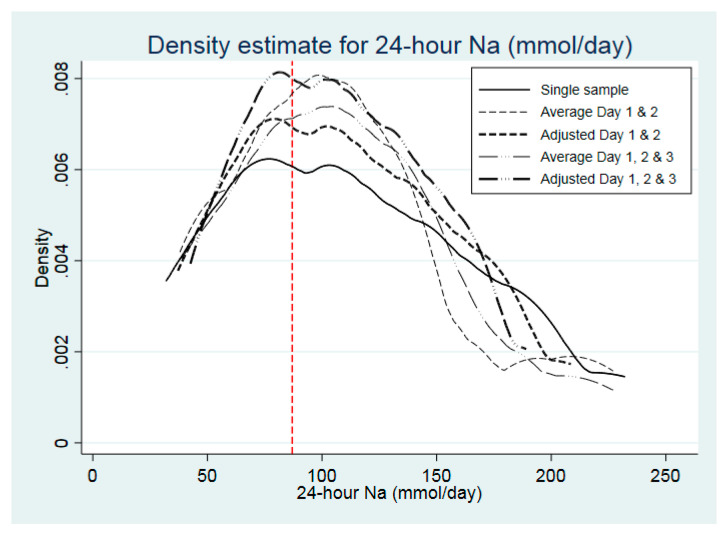
Population distribution of 24 h urinary Na (mmoL/day), according to number of days of urine collection (*n* = 23). Density refers to proportion of persons, e.g. 0.002 = 0.2%). Solid line: raw data from day1 only; thin dashed line: after correction for average of day1 and day2; thick dashed line after correction for intra-individual variation of day1 and day2; thin long dashed & dotted line: after correction for average of all three days; thick long dashed & dotted line after correction for intra-individual variation of all three days. Vertical dashed lines correspond to 5 g salt/day (equivalent to 87 mmoL/day).

**Table 1 nutrients-12-02026-t001:** Sample demographic and clinical characteristics.

Characteristics	All Subjects (*n* = 43)	Subjects with 3-Days Valid Measurements (*n* = 23)
Sex, *n* (%)		
Women	31 (72.1)	16 (69.6)
Men	12 (27.9)	7 (30.4)
Age (years)	62 (19)	58 (18)
BMI (Kg/m^2^)	30.1 (9.3)	33.8 (9.6)
Normal weight (18.5–24.9 kg/m^2^), *n* (%)	8 (18.6)	1 (4.3)
Overweight (25–30 kg/m^2^), *n* (%)	12 (27.9)	6 (26.1)
Obese (≥30 kg/m^2^), *n* (%)	23 (53.5)	16 (69.6)
Systolic BP (mm Hg)	140 (40)	133 (41)
Diastolic BP (mm Hg)	88 (27)	88 (26)

BMI: Body Mass Index, BP: Blood Pressure. For the total sample (*n* = 43) and for the sample with 3-days valid measurements (*n* = 23), data are presented as median (IQR), unless otherwise indicated.

**Table 2 nutrients-12-02026-t002:** Spot and 24 h urinary Na, K, and Cr, by collection day.

Characteristics	Day 1 (*n* = 23)	Day 2 (*n* = 23)	Day 3 (*n* = 23)
Spot Na (mmoL/L)	89.0 (59.0)	90.0 (52.0)	80.0 (70.0)
Spot K (mmoL/L)	26.7 (14.4)	32.0 (35.5)	22.0 (23.4)
Spot Cr (mmoL/L)	9.7 (8.5)	9.6 (6.8)	7.3 (5.3)
24 h urine volume (ml)	1100.0 (580.0)	1100.0 (995.0)	1200.0 (700.0)
24 h Na (mmoL/day)	107.0 (88.0)	95.0 (45.0)	80.0 (84.0)
24 h K (mmoL/day)	33.9 (22.5)	27.3 (24.1)	27.5 (21.8)
24 h Cr (mmoL/day)	9.9 (8.3)	8.3 (8.4)	9.9 (9.1)

Na: sodium, K: potassium, Cr: creatinine. For each day of collections and for each compound, data are presented as median (IQR).

**Table 3 nutrients-12-02026-t003:** Spot and 24 h Na urinary concentration distribution, raw data from day 1, after correction for intra-individual variation of two and three spot sample collections and averages of multiple collections.

	Raw Data for Day 1	Average of Days 1 and 2	Day 1 Corrected Using 2 Replicates	Average of Days 1, 2, and 3	Day 1 Corrected Using 3 Replicates
**Spot Na (mmoL/L)**					
Minimum	14.00	23.50	21.71	35.00	27.79
p5 ^a^	44.00	36.00	51.33	39.00	55.99
p10	45.00	54.50	52.21	49.00	56.76
p25	60.00	71.50	64.81	66.33	67.69
p50	89.00	92.00	87.16	85.33	86.14
p75	119.00	107.50	108.43	112.67	102.89
p90	163.00	136.00	137.36	150.67	124.73
p95	186.00	152.00	151.68	162.33	135.21
IQR	59.00	36.00	43.62	46.33	35.21
Maximum	193.00	186.50	155.95	179.00	138.30
Mean	93.30	93.35	88.36	92.86	86.19
SD	45.66	35.57	33.04	36.11	26.59
**24 h Na (mmoL/day)**					
Minimum	32.00	38.00	37.15	38.67	42.60
p5	49.00	42.50	53.84	41.67	58.70
p10	56.00	46.00	60.48	48.00	64.90
p25	66.00	71.00	69.78	68.67	73.44
p50	107.00	107.00	106.29	110.67	105.64
p75	154.00	130.50	145.95	141.33	138.93
p90	201.00	195.50	184.05	172.33	169.76
p95	220.00	202.50	199.11	213.00	181.69
IQR	88.00	59.50	76.17	72.67	65.49
Maximum	232.00	227.00	208.54	227.00	189.10
Mean	115.00	107.87	111.71	109.22	109.12
SD	56.33	50.77	48.09	50.54	40.92

^a^*p* = percentile. Na: sodium. The intra-individual variation (sb/sobs) was, respectively, for spot and 24 h Na: 0.752 and 0.871 when calculated using two replicates and 0.612 and 0.752 when calculated using three replicates.

**Table 4 nutrients-12-02026-t004:** Prediction equations to estimate 24 h Na using correction applied for both spot Na and spot Cr using the INTERSALT equation (mmoL/day).

Statistics	Measured 24 h Urinary Sodium Excretion	INTERSALT Spot 1	INTERSALT Mean Day 1 and 2	INTERSALT Adjusted Day 1 and 2	INTERSALT Mean Day 1, 2, and 3	INTERSALT Adjusted Day 1, 2, and 3
Minimum	32.00	4.48	12.06	11.00	14.21	12.27
p5	49.00	21.09	39.03	42.41	48.53	45.41
p10	56.00	90.52	102.67	105.63	109.21	106.67
p25	66.00	132.82	119.21	122.35	120.76	122.20
p50	107.00	140.74	132.96	141.44	133.02	141.39
p75	154.00	167.67	164.70	164.97	172.34	165.93
p90	201.00	212.32	209.14	208.46	198.68	204.26
p95	220.00	215.77	209.20	213.71	207.71	206.18
IQR	88.00	34.85	45.48	42.62	51.57	43.74
Maximum	232.00	220.26	215.19	214.36	228.70	213.41
Mean	115.00	140.87	140.17	142.80	140.58	142.15
SD	56.33	51.30	49.01	48.10	47.86	47.01
*p*-value *		0.0208	0.0150	0.0150	0.0177	0.0163
Spearman *r*		0.3083	0.2856	0.2727	0.2885	0.2579
*p*-value		0.1524	0.1865	0.2080	0.1818	0.2348

* *p*-value of difference between measured 24 h urinary sodium excretion and each prediction equation using INTERSALT was assessed by Wilcoxon’s signed rank test. Na: sodium, Cr: creatinine.

**Table 5 nutrients-12-02026-t005:** Prediction equations to estimate 24 h Na using correction applied for both spot Na and spot Cr using the Tanaka equation (mmoL/day).

Statistics	Measured 24 h Urinary Sodium Excretion	Tanaka Spot 1	Tanaka Mean Day 1 and 2	Tanaka Adjusted Day 1 and 2	Tanaka Mean Day 1, 2, and 3	Tanaka Adjusted Day 1, 2, and 3
Minimum	32.00	124.32	176.45	182.74	227.93	204.29
p5	49.00	207.34	225.89	231.07	244.95	236.30
p10	56.00	238.77	236.59	287.05	245.80	293.09
p25	66.00	285.11	283.61	325.21	276.53	332.74
p50	107.00	396.58	370.48	374.39	370.83	377.85
p75	154.00	482.93	415.18	438.87	431.25	419.99
p90	201.00	559.35	485.55	478.08	461.72	458.77
p95	220.00	572.75	500.06	498.03	477.71	478.86
IQR	88.00	197.82	131.56	113.66	154.72	87.25
Maximum	232.00	738.90	597.13	541.18	498.26	522.32
Mean	115.00	392.03	364.27	376.83	364.58	374.55
SD	56.33	138.88	97.35	84.72	83.76	74.80
*p*-value *		<0.0001	<0.0001	<0.0001	<0.0001	<0.0001
Spearman correlation coefficient		0.1047	0.3192	0.1808	0.2372	0.1927
*p*-value correlation		0.6343	0.1377	0.4090	0.2759	0.3784

* *p*-value of difference between measured 24 h urinary sodium excretion and each prediction equation using Tanaka was assessed by Wilcoxon’s signed rank test. Na: sodium, Cr: creatinine.

**Table 6 nutrients-12-02026-t006:** Prediction equations to estimate 24 h Na using correction applied for both spot Na and spot Cr using the Kawasaki equation (mmoL/day).

Statistics	Measured 24 h Urinary Sodium Excretion	Kawasaki Spot 1	Kawasaki Mean Day 1 and 2	Kawasaki Adjusted Day 1 and 2	Kawasaki Mean Day 1, 2, and 3	Kawasaki Adjusted Day 1, 2, and 3
Minimum	32.00	130.04	203.27	212.56	281.76	245.04
p5	49.00	288.74	341.67	331.53	357.15	341.13
p10	56.00	349.23	373.04	403.65	366.99	422.93
p25	66.00	445.93	428.95	524.98	415.92	526.37
p50	107.00	680.59	615.32	685.76	620.88	671.39
p75	154.00	913.44	838.75	824.38	834.91	791.88
p90	201.00	1033.24	939.30	907.04	990.56	904.10
p95	220.00	1060.30	940.61	961.78	1001.26	953.36
IQR	88.00	467.50	409.79	299.39	418.99	265.51
Maximum	232.00	1360.01	975.35	1102.78	1013.84	1048.92
Mean	115.00	693.41	633.74	660.08	635.66	654.93
SD	56.33	290.66	227.94	210.98	223.79	198.86
*p*-value *		<0.0001	<0.0001	<0.0001	<0.0001	<0.0001
Spearman correlation coefficient		0.1650	0.3192	0.2292	0.2806	0.2085
*p*-value correlation		0.4518	01377	0.2927	0.1946	0.3397

* *p*-value of difference between measured 24 h urinary sodium excretion and each prediction equation using Kawasaki was assessed by Wilcoxon’s signed rank test. Na: sodium, Cr: creatinine.

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
