# Peer review of "Correcting for Intra-Individual Variability in Sodium Excretion in Spot Urine Samples Does Not Improve the Ability to Predict 24 h Urinary Sodium Excretion"

_nutrients, 2020, doi:10.3390/nu12072026_

Round 1

Reviewer 1 Report

Thank you for the invitation to review this interesting paper which examines individual variability in sodium excretion in a relatively small sample of participants in the WHO-SAGE study in South Africa.   It adds to the growing body of literature that describes the serious limitations of spot urine collection to estimate individual or population sodium intake.  The novelty of this paper is its use of an adjustment equation to account for individual variability based on multiple days of urine collection (both 24 hour and spot).  I have a number of comments and suggestions below.

Introduction: Lines 47-48- 2300mg sodium is closer to 6g salt (5.8 grams) than 5 grams.  WHO recommends that adults consume less than 2000mg sodium.(World Health Organization 2012)

Methods:  Can you say why early morning urines were used rather than random urines in the methods section.

Can you comment on how the adjusted Na equation was derived- is this based on other research?  If so, could it be referenced here?

Results: Participant demographics- it would be good to include the age range- which looks like it is quite wide given the SD.  Also please indicate the ethnicity/race of participants.  This is likely to be important, especially given the differences demonstrated in previous studies (eg (Swanepoel, Schutte et al. 2018)). 

Figures 2 and 3 appear almost identical, other than the line indicating 5g of salt/day and the units in the x axis.  I wonder if they could be combined?

Table 4c- legend- are these units correct (mmol/L)? or should they be the same as in the legend for Table 4b (mmol/day)?

Tables 4a and 4b- are striking in the difference between estimates- especially for the Kawasaki equation at 6 times the estimated sodium excretion from 24 hour urinary excretion.  This is quite different to other research which suggests that while spot urine estimates under and over-estimate actual excretion at the ends of the range, mean levels are relatively close – for example (Huang, Crino et al. 2016).  I may have mis-interpreted the numbers in the tables (in which case I suggest they need further explanation).  Otherwise, if these figures are correct this is a finding at odds with much other research- including in South Africa(Swanepoel, Schutte et al. 2018), and needs highlighting. 

Discussion:  A number of studies have previously examined whether repeated spot urine collections (repeated early morning or random) improve accuracy of estimates of 24 hour urinary excretion.(Iwahori, Ueshima et al. 2014, Iwahori, Ueshima et al. 2016, Uechi, Asakura et al. 2016)  These could be usefully included in the discussion. 

Line 261- I imagine body size and composition may also contribute to conflicting findings in different ethnic groups.

Line 201-2  It would be useful to clarify what is meant by ‘multiple’.  Previous research has suggested that up to 10 repeated 24h urine samples may be required for an accurate estimate of usual sodium intake.(Cogswell, Elliott et al. 2013, Campbell, He et al. 2019)

References:

Campbell, N. R., F. J. He, M. Tan, F. P. Cappuccio, B. Neal, M. Woodward, M. E. Cogswell, R. McLean, J. Arcand and G. MacGregor (2019). "The International Consortium for Quality Research on Dietary Sodium/Salt (TRUE) position statement on the use of 24‐hour, spot, and short duration (< 24 hours) timed urine collections to assess dietary sodium intake." The Journal of Clinical Hypertension.

Cogswell, M. E., P. Elliott, C.-Y. Wang, D. G. Rhodes, C. M. Pfeiffer and C. M. Loria (2013). "Assessing U.S. Sodium Intake through Dietary Data and Urine Biomarkers." Advances in Nutrition: An International Review Journal 4(5): 560-562.

Huang, L., M. Crino, J. H. Y. Wu, M. Woodward, F. Barzi, M.-A. Land, R. McLean, J. Webster, B. Enkhtungalag and B. Neal (2016). "Mean population salt intake estimated from 24-h urine samples and spot urine samples: a systematic review and meta-analysis." International Journal of Epidemiology 45(1): 239-250.

Iwahori, T., H. Ueshima, N. Miyagawa, N. Ohgami, H. Yamashita, T. Ohkubo, Y. Murakami, T. Shiga and K. Miura (2014). "Six random specimens of daytime casual urine on different days are sufficient to estimate daily sodium/potassium ratio in comparison to 7-day 24-h urine collections." Hypertens Res 37(8): 765-771.

Iwahori, T., H. Ueshima, S. Torii, Y. Saito, A. Fujiyoshi, T. Ohkubo and K. Miura (2016). "Four to seven random casual urine specimens are sufficient to estimate 24-h urinary sodium/potassium ratio in individuals with high blood pressure." J Hum Hypertens 30(5): 328-334.

Swanepoel, B., A. E. Schutte, M. Cockeran, K. Steyn and E. Wentzel-Viljoen (2018). "Monitoring the South African population’s salt intake: spot urine v. 24 h urine." Public health nutrition 21(3): 480-488.

Uechi, K., K. Asakura, R. Yui, S. Masayasu and S. Sasaki (2016). "Advantage of multiple spot urine collections for estimating daily sodium excretion: comparison with two 24-h urine collections as reference." Journal of hypertension 34(2): 204-214.

World Health Organization (2012). Guideline: Sodium intake for adults and children. Geneva, World Health Organization (WHO).

Author Response

Introduction: Lines 47-48- 2300mg sodium is closer to 6g salt (5.8 grams) than 5 grams.  WHO recommends that adults consume less than 2000mg sodium.(World Health Organization 2012)

This has been corrected to 2000mg.

Methods:  Can you say why early morning urines were used rather than random urines in the methods section.

This has been explained as follows: “Early morning rather than random samples were collected for spot samples due to logistics of the fieldwork and because of the evidence that early morning samples more closely reflect diurnal variation.”

Can you comment on how the adjusted Na equation was derived- is this based on other research?  If so, could it be referenced here?

References 18 and 19 describe this method, and have been added to the methods section.

Results: Participant demographics- it would be good to include the age range- which looks like it is quite wide given the SD.  Also please indicate the ethnicity/race of participants.  This is likely to be important, especially given the differences demonstrated in previous studies (eg (Swanepoel, Schutte et al. 2018)). 

Additional information has been added.

These were all from the same ethnic background (African), geographical region (Gauteng province).

Figures 2 and 3 appear almost identical, other than the line indicating 5g of salt/day and the units in the x axis.  I wonder if they could be combined?

These figures have been merged and the super-imposed red dotted vertical line included to indicate equivalent for 5g salt (87 mmol/day).

Table 4c- legend- are these units correct (mmol/L)? or should they be the same as in the legend for Table 4b (mmol/day)?

This was a typo and has been corrected (mmol/day).

Tables 4a and 4b- are striking in the difference between estimates- especially for the Kawasaki equation at 6 times the estimated sodium excretion from 24 hour urinary excretion.  This is quite different to other research which suggests that while spot urine estimates under and over-estimate actual excretion at the ends of the range, mean levels are relatively close – for example (Huang, Crino et al. 2016).  I may have mis-interpreted the numbers in the tables (in which case I suggest they need further explanation).  Otherwise, if these figures are correct this is a finding at odds with much other research- including in South Africa(Swanepoel, Schutte et al. 2018), and needs highlighting. 

We were surprised to find such a stark magnitude of difference, however the data has been checked and is correct. Reasons as to differences with other published data, including another South African study by Swanepoel, Schutte et al. 2018 are possibly related to the older age of participants as well as the small sample size, or may be due to a more homogenous dietary intake, post mandatory sodium legislation in South Africa. It is noteworthy that only one study included in a meta analysis of the relationship between spot urinary Na and 24hr Na excretion, using these equations (Huang, Crino et al. 2016) included African subjects (PURE study, Mente et al. 2014).

Additional text has been added to the Discussion to further explain this finding, with suggested references added:

“The finding that the Kawasaki equation provided estimates at 6 times that of 24-hour urinary Na excretion is in contrast to other research [42] which suggests that while spot urine estimates under and over-estimate actual excretion at extremes of the range, mean levels are relatively close. Another study from South Africa reported a lower magnitude of error but similarly reported greatest bias with the Kawasaki equation and least with the INTERSALT equation [16].”

Discussion:  A number of studies have previously examined whether repeated spot urine collections (repeated early morning or random) improve accuracy of estimates of 24 hour urinary excretion.(Iwahori, Ueshima et al. 2014, Iwahori, Ueshima et al. 2016, Uechi, Asakura et al. 2016)  These could be usefully included in the discussion. 

Thankyou for this suggestion which has been added as follows:

“In our study, all spot collections were taken as early morning samples, as has been done in another validation study that found three repeated early morning spot samples most closely approximated 24hr urinary Na collections [37]. However, other studies have reported that higher numbers of repeated spot urine collections, ranging from four to seven collected at random, improved accuracy of estimates of repeated 24 hour urinary excretions.”

Line 261- I imagine body size and composition may also contribute to conflicting findings in different ethnic groups.

BMI was high at a median of 33.8 (9.6) kg/m2 and ranged  from 23.3 to 56.6. The prediction equations were developed and validated in Japanese and USA populations, not Africans therefore there could be errors in applying these equations to ethnic groups with a much larger prevalence of obesity. 

Text added as follows:

“Conflicting findings may thus be due to differences in dietary patterns and confounding factors associated with ethnicity, culture and cuisine as well as differences in body size and composition.”

and into conclusion: “Spot urine collections, proposed as a proxy for assessing 24hr Na excretion using prediction equations, are not valid for use in an urban older obese population in South Africa.”

Line 201-2  It would be useful to clarify what is meant by ‘multiple’.  Previous research has suggested that up to 10 repeated 24h urine samples may be required for an accurate estimate of usual sodium intake.(Cogswell, Elliott et al. 2013, Campbell, He et al. 2019).

This information has been included, thankyou for the suggestion to clarify the term “multiple.”

“Previous research has suggested that up to 10 repeated 24h urine samples may be required for an accurate estimate of usual sodium intake [46,47].”

Reviewer 2 Report

The study by Charlton et al., seeks to determine if multiple spot urines improves prediction of 24hr sodium intake compared to a single collection and further whether adjusting for intra-individual variation in urinary sodium excretion using repeated 24 hr collections affect daily estimate. Complete data on 23 older adults was included. Data highlights inaccuracy of spot urines that is not improved by additional collections. Better explanations under each table and figure would aid reader in understanding the data presented. Figures should be improved for clarity. Overall, this study confirms what we already know about spot urines.

Given that your results are only applicable to older adults, authors should consider including that in the title

Author list remove “and” between Wepener and Corso.

No email for Schutte. Schutte associated with HART but email listed for Wepener

Abstract

States 48 participated but results clearly state only 43 are presented and really most of the data is on 23.

Fix median age as well. Again, I think using a mean +/- SD is better here.

Intro

Line 82 change from “for each participant” to “from each participant”

Line 85 remove “has been applied” at very end of line. Already stated for dietary data and not necessary to repeat. “but less frequently to biochemical data” is cleaner

Line 87-88 “it contracted the distribution”. What do you mean by contracted?

Line 90. Change to “The objective of this study was to evaluate the impact of collecting …” and “comparing this with the effect …”

Methods

Line 130. Is there a reference to support the BP cut-offs?

Results

Line 157 Should be “and 1 man”

The removal of the 5 subjects should be addressed in methods.

Line 159 “had 3 days of valid measurements” remove all

Line 160 “… Cr values from spot and 24 hr urine, …”

Table 1. BP should be written in whole numbers, no decimals. BP is never given with decimal places to individuals. It’s not clear why data is presented as median. And this should be discussed in the statistical section of the methods

Given that sodium intake based on the 24 hr urinary sodium excretions is rather low (day 1: 2461; day 2: 2185; day 3: 1840), it seems that you are really limited in applicability of these subjects. A greater breadth of intake would have been beneficial. Older adults tend to eat less energy and therefore, less sodium. A great of young to middle-aged adults would be of benefit to study

Change column headings to “All Subjects (n=43)” and “Subjects with 3-days of valid measurements”

No reason to include a line for underweight BMI since no subjects met this BMI category

Table 2. Acronyms should be explained below table.

Text in figures looks odd. Hard to tell lines apart. Applies to all figures

A better explanation of how the data is represented in all tables and figures is warranted

Conclusion should be clear that this is applicable to older adults as average age was around 60 years

Author Response

The study by Charlton et al., seeks to determine if multiple spot urines improves prediction of 24hr sodium intake compared to a single collection and further whether adjusting for intra-individual variation in urinary sodium excretion using repeated 24 hr collections affect daily estimate. Complete data on 23 older adults was included. Data highlights inaccuracy of spot urines that is not improved by additional collections. Better explanations under each table and figure would aid reader in understanding the data presented. Figures should be improved for clarity. Overall, this study confirms what we already know about spot urines.

More information has been included in the footnotes on both tables and figures. Figures have been redrawn using different lines for better clarity. Additionally, EPS file versions will be provided to the journal for the purpose of publication.

Given that your results are only applicable to older adults, authors should consider including that in the title.

We prefer not to limit the title but have included the words “older adults” in the conclusion.  

Author list remove “and” between Wepener and Corso.

Done

No email for Schutte. Schutte associated with HART but email listed for Wepener

Abstract

States 48 participated but results clearly state only 43 are presented and really most of the data is on 23.

We present the full figure of n = 48 to demonstrate difficulty in obtaining three valid 24hr urinary collections, but as the reviewer correctly states data is analysed only for n = 23, as indicated throughout the manuscript.

This has been changed to n = 23 in abstract.

Mean (SD) age is provided.  

Intro

Line 82 change from “for each participant” to “from each participant”

Line 85 remove “has been applied” at very end of line. Already stated for dietary data and not necessary to repeat. “but less frequently to biochemical data” is cleaner

Line 87-88 “it contracted the distribution”. What do you mean by contracted?

Text replaced with “it narrowed extremes of the distribution”

Line 90. Change to “The objective of this study was to evaluate the impact of collecting …” and “comparing this with the effect …”

This has been done.

Methods

Line 130. Is there a reference to support the BP cut-offs?

We refer to our previous published study

Results

Line 157 Should be “and 1 man”.  The removal of the 5 subjects should be addressed in methods.

The first sentence has been changed to reflect all exclusions as it was confusing. “Of the 48 participants who collected repeated urine samples, only 23 had three days of valid collections and were thus included in the analysis.”

 Line 159 “had 3 days of valid measurements” remove all

This has been done.

 Line 160 “… Cr values from spot and 24 hr urine, …”

This has been done.

Table 1. BP should be written in whole numbers, no decimals. BP is never given with decimal places to individuals. It’s not clear why data is presented as median. And this should be discussed in the statistical section of the methods.

BP rounded off. Data was not normally distributed and therefore presented as median (IQR) as is the statistical convention.

Sentence added to results as follows: “Descriptive data is presented as median (IQR) because of non normally distributions.”

Given that sodium intake based on the 24 hr urinary sodium excretions is rather low (day 1: 2461; day 2: 2185; day 3: 1840), it seems that you are really limited in applicability of these subjects. A greater breadth of intake would have been beneficial. Older adults tend to eat less energy and therefore, less sodium. A great of young to middle-aged adults would be of benefit to study.

Thank you for this comment. We acknowledge these aspects of generalizability in the last sentence of the discussion as follows: “Given the older age group and overall low 24hr Na excretion, generalizability to wider age groups that may have more diverse dietary patterns is limited.”

Change column headings to “All Subjects (n=43)” and “Subjects with 3-days of valid measurements”

This has been done.

No reason to include a line for underweight BMI since no subjects met this BMI category.

This has been excluded.

Table 2. Acronyms should be explained below table.

A better explanation of how the data is represented in all tables and figures has been included.  

Conclusion should be clear that this is applicable to older adults as average age was around 60 years

This has been included as follows:
“Spot urine collections, proposed as a proxy for assessing 24hr Na excretion using prediction equations, are not valid for use in an urban older obese population in South Africa.”

Round 2

Reviewer 1 Report

The authors have revised their paper, and it is suitable for publication.  It will make a valuable contribution to the literature.